# The development of blood protein profiles in extremely preterm infants follows a stereotypic evolution pattern

Wen Zhong [1,2,13], Hanna Danielsson [3,4,13], Nele Brusselaers [3,5], Dirk Wackernagel[6], Ulrika Sjöbom[7,8], Karin Sävman[9,10], Ingrid Hansen Pupp[11], David Ley[11], Anders K. Nilsson [7], Linn Fagerberg [12], Mathias Uhlén [2,12] & Ann Hellström [7✉]

## Abstract

**Background** Preterm birth is the leading cause of neonatal mortality and morbidity. Early diagnosis and interventions are critical to improving the clinical outcomes of extremely premature infants. Blood protein profiling during the first months of life in preterm infants can shed light on the role of early extrauterine development and provide an increased understanding of maturation after extremely preterm birth and the underlying mechanisms of prematurity-related disorders.

**Methods** We have investigated the blood protein profiles during the first months of life in preterm infants on the role of early extrauterine development. The blood protein levels were analyzed using next generation blood profiling on 1335 serum samples, collected longitudinally at nine time points from birth to full-term from 182 extremely preterm infants.

**Results** The protein analysis reveals evident predestined serum evolution patterns common for all included infants. The majority of the variations in blood protein expression are associated with the postnatal age of the preterm infants rather than any other factors. There is a uniform protein pattern on postnatal day 1 and after 30 weeks postmenstrual age (PMA), independent of gestational age (GA). However, during the first month of life, GA had a significant impact on protein variability.

**Conclusions** The unified pattern of protein development for all included infants suggests an age-dependent stereotypic development of blood proteins after birth. This knowledge should be considered in neonatal settings and might alter the clinical approach within neonatology, where PMA is today the most dominant age variable.

### Plain language summary

Being born too early can affect a baby's health. We looked at how babies born extremely preterm, meaning more than 12 weeks earlier than a full-term baby, develop. We looked at the proteins present in their blood from the day they were born until their original due date. Our study of 182 extremely preterm babies born at different points in the pregnancy (gestational ages) found that the proteins present in their blood changed in a similar way over time. This means that the age of a baby after birth, and not how early they were born, mostly affects the proteins in their blood. These findings help us understand how extremely preterm babies develop after birth, which could lead to improvements to their healthcare during the first few weeks of their life.

[1] Science for Life Laboratory, Department of Biomedical and Clinical Sciences (BKV), Linköping University, Linköping, Sweden. [2] Department of Neuroscience, Karolinska Institutet, Stockholm, Sweden. [3] Centre for Translational Microbiome Research, Department of Microbiology, Tumor and Cell Biology, Karolinska Institutet, Stockholm, Sweden. [4] Sach's Children's and Youth Hospital, Södersjukhuset, Stockholm, Sweden. [5] Global Health Institute, Antwerp University, Antwerp, Belgium. [6] Department of Neonatology, Karolinska University Hospital and Institute, Astrid Lindgrens Children's Hospital, Stockholm, Sweden. [7] Department of Clinical Neuroscience, Institute of Neuroscience and Physiology, Sahlgrenska Academy, University of Gothenburg, Gothenburg, Sweden. [8] Learning and Leadership for Health Care Professionals At the Institute of Health and Care Science at Sahlgrenska Academy at University of Gothenburg, Gothenburg, Sweden. [9] Department of Pediatrics, Institute of Clinical Sciences, Sahlgrenska Academy, University of Gothenburg, Gothenburg, Sweden. [10] Region Västra Götaland, Dept of Neonatology, The Queen Silvia Children's Hospital, Sahlgrenska University Hospital, Gothenburg, Sweden. [11] Department of Pediatrics, Institute of Clinical Sciences Lund, Lund University and Skane University Hospital, Lund, Sweden. [12] Science for Life Laboratory, Department of Protein Science, KTH—Royal Institute of Technology, Stockholm, Sweden. [13] These authors contributed equally: Wen Zhong, Hanna Danielsson. ✉email: ann. hellstrom@medfak.gu.se

Preterm birth is the leading cause of neonatal morbidity and mortality and a significant risk factor for long-term neurodevelopmental and respiratory impairment[1–3]. The number of births before 28 weeks of gestation, defined as extremely preterm, was estimated to be approximately 600,000 in 2014 worldwide[4]. Although significant advances in neonatal intensive care over the last few decades resulted in increased survival rates[4–7], extremely preterm birth is associated with a high rate of neonatal and childhood diseases[8].

The etiology behind preterm morbidities is multifactorial and poorly understood. Infants born before 28 gestational weeks spend the third trimester outside the womb in an environment to which the newborn is not adapted. Investigations into the developmental trajectories, based on electronic health records (EHRs)[9] and high-throughput molecular profiling[10], can aid in identifying biomarkers for disease and diagnosis, where the information provided may serve as a basis for personalized medicine[11]. Analyzing the evolution of blood protein profiles can provide valuable insights into the postnatal development of these immature infants[12,13]. Knowledge about the postnatal proteome evolution in relation to postmenstrual age (PMA) and postnatal age (PNA), respectively, can shed light on the role of early extrauterine development in health and disease. Thus, there is an urgent need to understand the patterns and dynamics of blood protein profiles during the early development of preterm infants.

The proximity extension assay (PEA) technology developed by Olink, also referred to as "next generation blood profiling", allows for the analysis of blood levels of hundreds and even thousands of proteins simultaneously in a small blood volume without sacrificing accuracy or sensitivity[12,14–16]. Immaturity and excessive blood sampling may increase the risk of anemia in preterm infants, and treatment with blood transfusions is associated with neonatal morbidities and impaired neurological development[17–19]. Thus, the assay platform is ideal for comprehensive proteome analysis in minute quantities of blood.

Longitudinal proteomic studies in preterm infants are scarce. A recent pilot study on 14 infants born at GAs 22 to 27 weeks used panels that targeted a large number of proteins based on the PEA principle to investigate the serum protein profiles[12]. The study showed dramatic changes in protein profiles unrelated to GA from day one through the early weeks of life. The extremely preterm infants had a distinct unified protein profile in serum on the first day after birth. Lee et al. explored dynamic proteome changes during the first week of life in 30 healthy term infants, revealing that these changes followed a robust developmental trajectory[20]. Furthermore, Olin et al. presented significant changes in protein expression by comparing cord blood from preterm and term infants with peripheral blood at later ages[21]. In a small cohort of term infants, age was the most prominent factor influencing serum protein expression in the first three years of life[22]. Nonetheless, the influence of prematurity on proteome development during the first weeks of life remains unresolved.

In the present study, we use the PEA method to investigate longitudinal serum protein profiles from birth to term-equivalent age (postmenstrual week (PMW) 40) and relate the results to GA, PMA, PNA, sex, and mode of delivery in a large cohort of extremely preterm infants. Thus, our results fill a critical knowledge gap on the evolution of the protein profiles in extremely preterm infants from premature birth to term-equivalent age. We identified a postnatal time-dependent stereotypic development of blood proteins. A developmental proteomics reference map for extremely preterm infants has been established to facilitate future pediatric research. It pinpoints a period where the degree of immaturity seems to significantly impact the proteome, thus directing a window for potential diagnostics and therapeutics.

## Methods

**Patients and nutritional management.** The current study is based on the multicenter, open-label, randomized controlled trial MegaDonnaMega (Clinical Trial.gov identifier NCT03201588). Details of the MegaDonnaMega-study are described elsewhere[23]. In summary, infants born before 28 weeks of gestation (ultrasonography dating) and treated at the neonatal intensive care unit (NICU) in Gothenburg, Lund, or Stockholm, Sweden, between December 2016 and December 2019, were randomized to receive the triglyceride oil supplement Formulaid® (DSM Nutritional Products Inc) containing arachidonic acid and docosahexaenoic acid or no extra supplement/standard care. Randomization was stratified according to the center and three GA groups: less than 25 weeks, 25 to 26 weeks, or 27 weeks. Twins or triplets were randomized to the same group. The lipid supplement was administrated enterally with a daily dose starting within 72 h of age and lasted until term-equivalent age. Mother's own milk (when available) or donor milk was the only enteral feed until a postmenstrual age of 33 weeks. Thereafter, donor milk was replaced with preterm formula. The enteral feeds were introduced, if possible, from the first day of life. All infants received at least some parenteral nutrition in the neonatal period. Details of the nutritional strategy and definitions of perinatal morbidities have been published previously[23,24] Written informed consent was obtained from the parents or guardians before inclusion.

The Mega Donna Mega study followed the Consolidated Standards of Reporting Trials (CONSORT) reporting guideline[25]. The regional ethical board of Gothenburg approved the Mega Donna Mega-study, and the Swedish Ethical Review Authority approved this extended study. The current study cohort included all infants in Mega Donna Mega with available longitudinal blood samples up to full-term age[23]. Standardized weight was calculated based on growth charts by Fenton and Kim[26].

**Sample and medical data collection.** Blood samples were taken in coordination with clinical routine samples on postnatal days 3, 7, 14, 28, followed by PMW 30, 32, 36, and term-equivalent age corresponding to 40 weeks PMA. At each sampling occasion, 0.6 ml blood was collected in a serum-separating tube with clot activator. Samples were kept refrigerated for a minimum of 30 min and a maximum of 2 h before centrifugation at RT 10 min $1500 \times g$. The sera were collected into polypropylene tubes and stored for up to one week at $-20\,°C$ before long-term storage at $-80\,°C$. The handling process can also be described with the SPREC code SER–SST–B–B–N–A–S. All longitudinal samples from one individual were allocated on the same plate, but randomized within the plate, to minimize batch-effects when studying within-individual protein expression patterns. Infants were randomized between plates based on center. Clinical data regarding birth, growth, nutrition and perinatal morbidities were collected prospectively according to the study protocol.

**Serum protein profiling.** Serum samples were gently thawed on ice, centrifuged at $4\,°C$ 20 min $1500 \times g$, and 25 μl serum was transferred to 96-well microtiter plates. Plates (17 in total) were deep-frozen and sent on dry ice to Olink Bioscience (Uppsala, Sweden) for analysis. Serum proteins were analyzed using a multiplex proximity extension assay (PEA) technology (Olink Bioscience) as previously described[12,15]. Briefly, each kit consists of a microtiter plate for measuring 92 protein biomarkers in all 88 samples, and each well contained 96 pairs of DNA-labeled antibody probes. Longitudinal samples from each individual were allocated to the same plate to reduce batch-effects related to inter-individual variability. To minimize inter- and intra-run variation, the data were normalized using both an internal control

(extension control) and an inter-plate control and then transformed using a pre-determined correction factor. This study uses six Olink panels including Cardiometabolic, Cardiovascular II, Cardiovascular III, Development, Inflammation, and Metabolism, resulting in 552 protein assays and 541 unique proteins. One microliter infant serum was use for each panel. The pre-processed data were provided in the arbitrary unit normalized protein expression (NPX) on a log2 scale, where a high NPX represents high protein concentration. Limit of detection (LOD) for each protein was defined as three standard deviations above the background. Protein panels from samples with more than 10% below LOD values were removed from the analysis. A preterm infant serum pool sample and 8 internal control samples were included on each plate for bridging and quality control. Three proteins with drastic fluctuations between visits were considered to have a problematic batch effect and were removed. After quality control, 538 unique proteins from 1335 samples were kept.

**Clustering analysis.** For hierarchical clustering, the NPX values of each protein were first standardized by scaling with a standard deviation of 1 and centered at 0. The scaled values from all 1335 samples were used to create the Euclidean distance matrix for dendrogram generation. Dendrograms showing gene expression in heatmaps have been clustered using the Ward2 algorithm, an implementation of "Ward's" minimum variance method implemented as "Ward.D2" in R package pheatmap[27]. UMAP analysis has been performed on NPX values of samples by using the R packages umap with default parameters. The diffusion map analysis was performed using the R package destiny[28] and the principal component analysis was performed using the prcomp function in the basic R package stats.

**Time-series expression analysis.** Mixed-effect modeling was performed using the lme4 package[29], and Kenward-Roger approximation[30] was used to calculate *P*-values with sex, delivery mode, subject and GA at birth as confounding factors. *P*-values were subsequently adjusted for multiple testing based on false discovery rate and considered significant if less than 0.01. In total, 451 differentially expressed proteins across nine visits were identified. The average NPX of each protein per visit was used for the hierarchical clustering analysis to create the dendrogram based on Euclidean distance. Ward's minimum variance method implemented as "Ward.D2" in the hclust function in the R package stats, where clusters are chosen at each stage such that the increase in cluster variance is minimized after merging. As a result, eight clusters were identified. For regression analysis, the NPX values from each protein in the particular cluster were standardized by scaling with a standard deviation of 1 centered at 0. Non-parametric local weighted regression (LOESS) was applied to generate the regression curve. Gene Ontology term (Biological Process) and BioPlanet pathway enrichment analyses for proteins in different time-series clusters were conducted using EnrichR[31] based on Fisher exact test.

**Prediction of postnatal age (PNA).** Generalized linear models with an elastic-net penalty was performed using glmnet[32] package in R. Specifically, the alpha value was set to 0.5, and 100 lambda values were tested. The "lambda.min" value, which is a measure of shrinkage, was determined after conducting a tenfold cross-validation analysis. Two-thirds (*n* = 934) of the samples were used for training the model, and the remaining samples (*n* = 401) were used as a validation.

**Statistical analysis and visualization.** In instances when infant postnatal age 4 weeks (PNW 4) occurred before PMW30, i.e., in infants born <26 weeks' GA, the PNW4 sample data was used for visualization and clustering. In all other analyses, the continuous chronological age (postnatal age or postmenstrual age) was used. Data analysis and visualization were performed using the R project[33] with the tidyverse suite of R packages[34] and the ggplot2 R package[35]. Variance analysis of the protein levels was conducted using multiple linear regression model with all protein significantly associated PNA, GA, sex, and mode of delivery as variables in the model. The fraction of explained variability was measured using Eta-squared ($\eta^2$), representing the proportion of the total sum of square explained by the factor, and was determined using analysis of variance (ANOVA) method with the build-in R function anova(). Differential expression analysis was conducted using ANOVA. The false discovery rate (FDR) was calculated using the p.adjust() function in R, which uses the Benjamini-Hochberg method. Proteins with FDRs less than 0.01 were considered differentially expressed proteins. Radar chart was generated using a R package fsmb[36].

**Reporting summary.** Further information on research design is available in the Nature Portfolio Reporting Summary linked to this article.

## Results

**The study cohort.** A total of 182 extremely preterm infants born before 28 weeks of gestation from the Mega Donna Mega study were selected based on the availability of longitudinal serum samples (Fig. 1a)[23]. Altogether 105 boys and 77 girls were included. Serum samples were collected repeatedly at nine planned time points (visits) from birth to term-equivalent age. Forty (22%) infants had nine complete samples, and 165 (91%) infants had at least six samples. Among the 182 infants included in the study, 177 (97.3%) survived to 40 weeks PMA. The enrolled infants were classified into three groups depending on GA at birth: group 1, born at less than 25 + 0 (weeks + days) (*N* = 61); group 2, born at 25 + 0 to 26 + 6 (weeks + days) (*N* = 81); and group 3, born at 27 + 0 to 27 + 6 (weeks + days) of gestation (*N* = 40) (Fig. 1b). The birth weights varied from 425 to 1345 grams (Supplementary Fig. 1a). Almost all (98.4%) of the infants had a birthweight appropriate for gestational age with standard deviations (SDS) > −2 (Supplementary Fig. 1b)[26]. Vaginal delivery was more common among infants born at lower gestational age (Mann–Whitney *U*-test, *P* = 0.017). We observed no significant differences in sex distribution or postnatal growth between GA groups (Fig. 1b and Supplementary Fig. 1b–d). The clinical characteristics of the three GA groups were summarized in Supplementary Data 1.

**Proteome profiling demonstrated dynamic changes in blood proteins after birth.** We analyzed 538 unique protein targets measured by six Olink PEA panels, including cardiometabolic, cardiovascular II and III, development, inflammation, and metabolism, for all 1335 collected blood serum samples (Supplementary Data 2). Protein levels measured as NPX were determined for each target and sample. An example of the protein expression determined by the Olink PEA technique can be seen in Fig. 2a, where the levels of fibroblast growth factor 21 (FGF-21) is shown from birth to full-term age (PMW 40). FGF-21 is a protein involved in metabolism and growth by regulating insulin sensitivity and glucose uptake[37]. We observed higher levels with increasing postnatal age of the infant (Fig. 2a), which is consistent with our previous report that FGF-21 serum levels were elevated

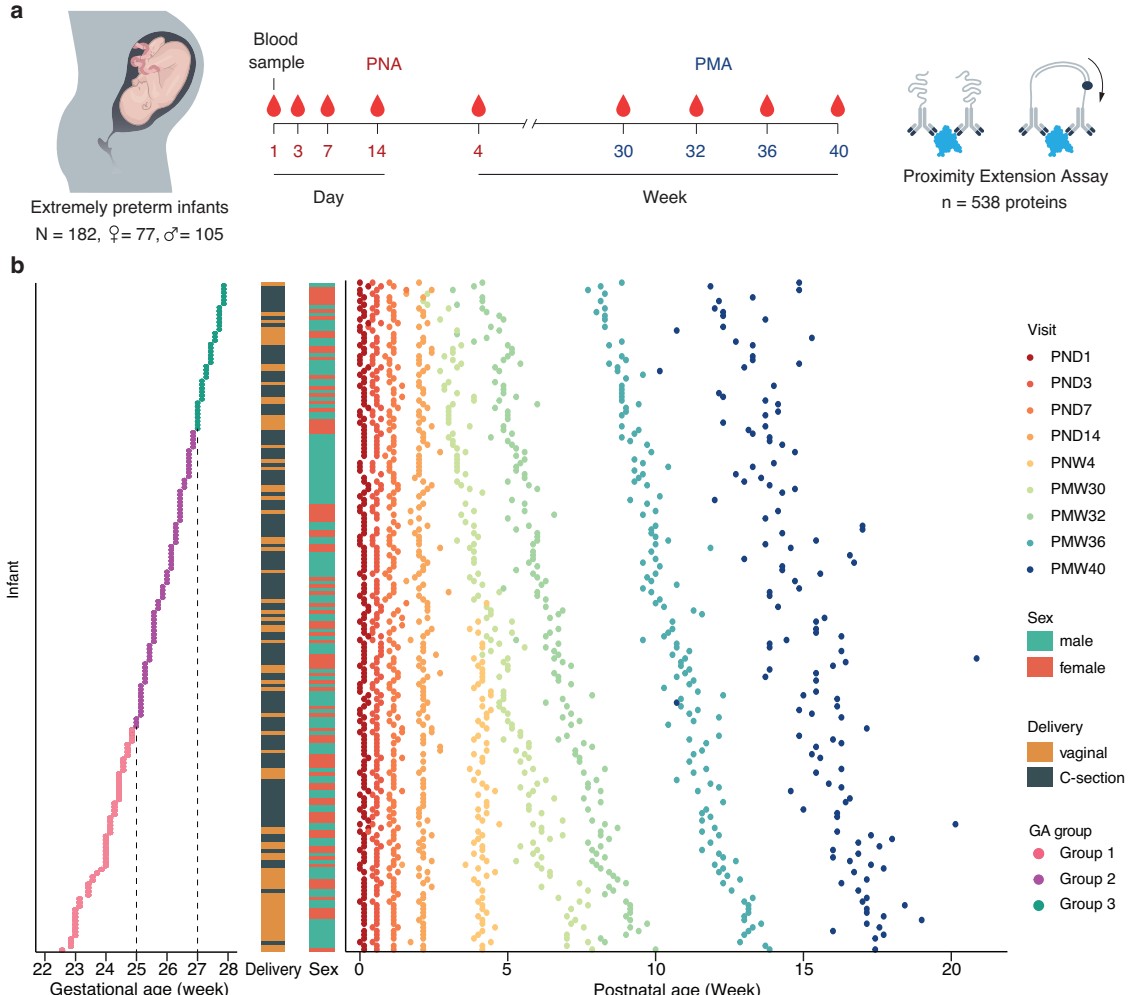

**Fig. 1 Overview of the study. a** 182 extremely preterm infants born before 28 weeks of gestation were enrolled in the study with repeated blood sampling from postnatal day (PND) 1 to postmenstrual week (PMW) 40. Using proximity extension assay, 538 unique protein targets were analyzed. **b** The infants were classified into three groups based on gestational age: group 1 (red), born at less than 25 + 0 (weeks + days) (N = 61); group 2 (purple), born at 25 + 0 to 26 + 6 (weeks + days) (N = 81); and group 3 (green), born at 27 + 0 to 27 + 6 (weeks + days) of gestation (N = 40). The sex and delivery mode distributions of the preterm infants are present in the bar plots. During the study period, altogether, 1335 blood samples were collected. Each infant is shown on the y-axis starting from the most immature (furthest to the bottom) and then ranked by gestational age (GA). Each dot along the same y-axis represents a unique sample from the same individual, color-coded by study visits.

after birth[12]. No differences in FGF-21 expression between sexes could be seen.

To explore variance of protein levels, inter-individual and intra-individual variations were calculated for each protein across all 182 infants and nine visits (Fig. 2b and Supplementary Data 3). Most of the proteins were observed with considerable variability in both, and FGF-21 was the most variable protein in the analysis. To get a comprehensive overview of the postnatal changes in the blood proteins, we analyzed the time-course expression patterns of the variable blood proteins from birth to 40 weeks PMA. Over time, differentially expressed proteins (451 out of 538) were identified using linear mixed-effect modeling with Benjamini-Hochberg adjusted *P*-value < 0.01, including 196 up-regulated proteins and 255 down-regulated proteins across the study visits (Fig. 2c). The longitudinal changes in protein expressions for each differentially expressed protein can be seen in the circular heatmap in Fig. 2d. Unsupervised hierarchical clustering analysis was further performed on the longitudinal expression profiles of the proteins based on Pearson correlation. A total of eight separate clusters, ranging in size from 34 to 84 proteins, were identified with variable time-course patterns (Figs. 2d, e and

Supplementary Data 4). As seen in Fig. 2e and Supplementary Fig. 2a, b, five clusters display overall declining protein levels and three increasing trends. The proteins most strongly changed over time included leptin (LEP), LDL-receptor, and several placenta elevated proteins, including Fc fragment of IgG receptor IIa (FCGR2A) and CGA (Supplementary Fig. 2c). The effect of GA at birth on the clustering trends was further explored. As seen in Supplementary Figs. 3a, b, protein levels in the three GA groups were almost the same at the birth, indicating similar expression patterns of proteins in different GA groups at birth.

**Functional analysis of the protein clusters**. The functions of the proteins in each of the eight identified clusters were explored. Tissue specificity of the proteins in the clusters was analyzed based on the Human Protein Atlas (HPA) classification[38,39]. This classification, elsewhere described, considers the level of gene expression in each tissue to determine the degree of specificity. Of the 451 proteins in the eight clusters, 101 proteins (22%), were annotated as tissue-enriched proteins according to the HPA classification (Fig. 3a and Supplementary Data 5). The analysis

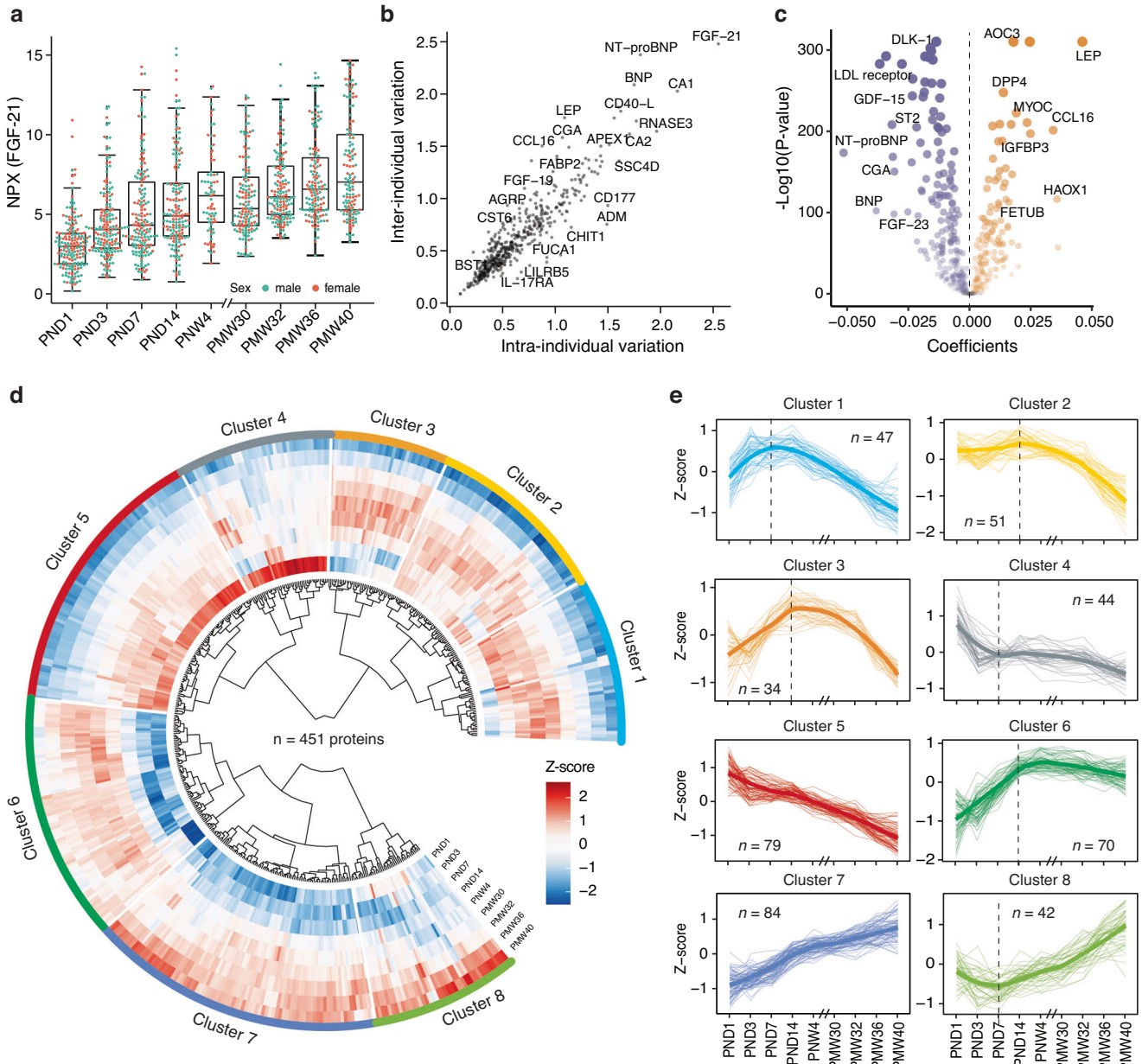

**Fig. 2 Time-variable expression profiles of blood proteins after birth. a** An example of the individual variability in protein expression during the study period from birth to full-term equivalent age, as shown for protein FGF-21. Box plots show medians and the 25th and 75th percentiles, whiskers show the largest and smallest values ($n = 1335$ samples), error bars represent mean ± SD, color-coded by sex. Each dot on the box-plot respresents an individual. **b** The variation of protein levels was calculated as the mean standard deviation (SD) for each protein within each visit and across all analyzed individuals ($n = 182$) (inter-individual) and as the mean SD for each protein within each individual across all sampling time points ($n = 9$) (intra-individual), respectively (see full details in Supplementary Data 3). **c** Volcano plot showing the differentially expressed proteins ($n = 451$) across the study visits. The x-axis represents the coefficients of the study visits. The y-axis represents −log10 adjusted P-values. Differentially expressed proteins were defined as proteins with adjusted P-values < 0.01 (mixed-effect modeling with sex, delivery mode, and gestational age (GA) as covariates). Multiple test corrections have been applied to the P-values using the Benjamini and Hochberg method. **d** Heatmap presenting the dynamic protein expression levels calculated as Z-scores of 451 differentially expressed proteins from birth to full-term, clustered based on the correlation between their expression profiles. **e** Eight clusters represent different trends in protein variation across the nine visits. The colored bold line represents the regression line based on all proteins and the total number of proteins in each cluster is also indicated in the plot.

showed that most of the liver, lymphoid tissue, and salivary gland enriched proteins were increased after birth, indicating the development of hepatic functions and immune and metabolic shifts during the neonatal period. Two examples include carboxylesterase 1 (CES1), a primary liver enzyme that functions in liver drug clearance[40], and Fc fragment of IgE receptor II (FCER2), which has essential roles in B-cell growth and

differentiation, as well as the regulation of IgE production (Fig. 3b)[41]. Many proteins that decreased after preterm birth were associated with the placenta, pancreas, and bone marrow, consistent with our previous findings (Fig. 3a)[12]. For example, Carboxypeptidase A1 (CPA1), a pancreas-enriched protein, is produced in the pancreas and preferentially cleaves C-terminal branched-chain and aromatic amino acids from dietary protein[42].

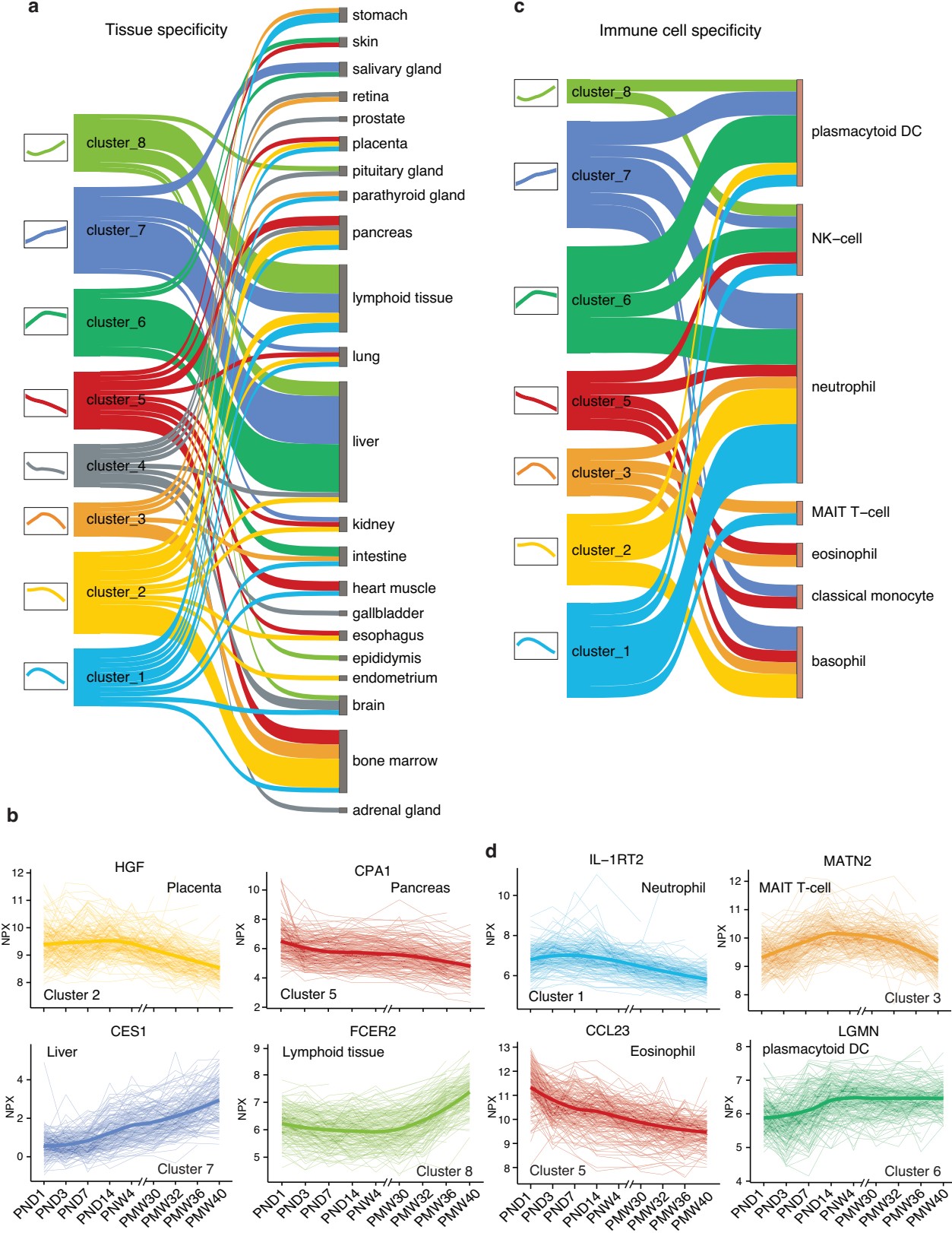

**Fig. 3 Tissue and immune cell specificity in relation to the protein clusters. a** Sankey diagram showing the numbers of proteins classified as tissue enriched in a specific tissue (n = 109) for each cluster. **b** Four examples of longitudinal expression patterns of tissue-enriched proteins for all 182 preterm infants with each individual connected with a line. The *y*-axis represents the NPX. The *x*-axis represents the blood sampling time points. The colors are coded by clusters. **c** Sankey diagram showing the numbers of proteins associated with specific immune cell types for each cluster. **d** Four examples of expression patterns of immune cell-enriched proteins.

Hepatocyte growth factor (HGF) is an acidic protein with a strong mitogenic effect on hepatocytes. Still, it is also enriched in the placenta with strong expression in the villous syncytium, extravillous trophoblast, and amnionic epithelium (Fig. 3b)[43,44].

To explore the postnatal development of the immune system in preterm infants, we investigated the immune cell specificity of the proteins with longitudinal dynamic changes. The cellular specificity is determined based on the gene expression levels in the 18 different immune cell types from the HPA (Fig. 3c, d)[39]. Here, 43 (9.5%) proteins were annotated as immune cell type enriched according to the HPA classification, which were at least four-fold higher expressed in one cell type than all other cell types (Supplementary Data 6). A large fraction of the proteins in clusters 6, 7, and 8 with increasing levels after birth are enriched in plasmacytoid dendritic cells. In contrast, most proteins with decreasing trends (clusters 1, 2, and 5) are enriched in basophils or neutrophils. The proteins in cluster 3 with an increasing trend followed by decreased protein levels have a mixed immune cell origin, including proteins enriched in T-cells, eosinophils, neutrophils, and basophils. This suggests that the expression activity of the dendritic cells increases after birth in these preterm infants, while the expression of proteins from basophils and neutrophils decreases.

In addition, functional enrichment analyses were performed to explore the modulated pathways for each identified cluster (Supplementary Fig. 4 and Supplementary Data 7a, b). As expected, multiple immune-related pathways were activated after preterm birth, including the T cell activation, inflammatory response pathway, interleukin-17 signaling pathway, and hematopoiesis pathway. Interestingly, the receptor for the advanced glycation end products (RAGE) pathway, which plays a vital role in leukocyte recruitment and have relatively high blood levels in extremely premature infants[45], was deactivated during postnatal development.

**Variance analysis of blood protein profiling after birth**. To assess how clinical aspects affect the blood protein expression levels, we established a linear regression model for each protein target and included four factors: PNA (days), GA at birth (days), sex, and mode of delivery (see Fig. 4a, b and Supplementary Data 8). The model revealed that the protein expression variations were primarily associated with PNA. This indicates that regarding protein variation, postnatal time is dominant compared to gestational age at birth. Nonetheless, GA at birth is the second most explanatory factor. Sex and delivery mode impacted a few specific proteins (Supplementary Figs. 5 and 6); however, overall substantially less influential compared to postnatal age. The contribution of the predictor variables was summarized in Supplementary Data 8. The effects of the maternal comorbidity (preeclampsia), the fatty acid supplementation and the percentage of mother's breast milk on protein expressions were also observed but less prominent when compared to those of PNA and GA (Supplementary Fig. 7). Moreover, to further expand the evaluation of impact of PNA on blood proteome expression, we investigate the potential of blood proteome as a predictor for PNA. We employed generalized linear models with an elastic-net penalty and identified a 'blood proteomic clock' comprising 151 proteins (Supplementary Data 9). The predictive PNA had a high level of consistency with chronological age, as demonstrated by a Pearson correlation coefficient of 0.98, and 95% of all samples were within ±1.96 SD range (13 days) according to a Bland–Altman plot (Fig. 4c and Supplementary Fig. 8). This suggests that the blood proteins are reliable measurements for estimating PNA of preterm infants. Interestingly, it was observed that among the infant samples that were of lower predicted PNA than actual PNA, the weight gain tended to be lower than normal (Supplementary Fig. 9).

The proteins with the highest effect within each analyzed factor have been highlighted in Fig. 4d–g. The influence of PNA was most prominent in the delta-like non-canonical Notch ligand 1 (DLK-1), with 75.1% of the serum protein level variance explained by PNA (Fig. 4d). DLK-1, also known as preadipocyte factor 1 (PREF1), is a marker of preadipocytes and inhibits adipogenesis[46]. It has been proposed that its function is to shift metabolism from lipid storage to peripheral lipid oxidation and act as a mediator of metabolic adaptation in early life[47]. DLK-1 levels were constant up to 30 weeks PMA, whereas after they decreased considerably, as shown in Fig. 4d.

As mentioned above, PNA seems more important than GA at birth in determining protein variance; however, for some proteins, variance is more associated with the GA at birth (seen to the right in Fig. 4a). One example was the tissue factor pathway inhibitor (TFPI), the primary inhibitor of the extrinsic coagulation pathway[48]. As illustrated in Fig. 4e, apparent differences in protein expression for TFP1 were observed between the three GA groups. The infants born at younger gestational ages had a persisting higher expression until full-term, when the expression levels converged. Furthermore, we show that the glycoprotein hormones alpha chain (CGA) levels decreased rapidly during the first days after birth and with a more significant decline in males, resulting in longitudinally lower levels in the male infants (Fig. 4f). CGA is one of the subunits that form the hormones human chorionic gonadotropin (hCG), luteinizing hormone (LH), follicle-stimulating hormone (FSH), and thyroid-stimulating hormone (TSH)[49]. Several proteins related to the mode of delivery were identified (Fig. 4a). The strongest association was seen for surfactant protein D (PSP-D, also called SP-D). An elevated level of PSP-D was observed in infants delivered by cesarean section (Fig. 4g).

**Distinct and coherent evolution of blood protein profiles over time**. To investigate the global molecular dynamics of preterm infants, we performed several dimensional reduction analyses, including Uniform Manifold Approximation and Projection (UMAP), based on the longitudinal protein expression profiles of all 182 infants (Fig. 5a and Supplementary Fig. 10a, b). Visualizing all 1335 samples the UMAP results revealed a distinct and stereotypic evolution of blood protein profiles from birth to term-equivalent age. The majority of the infants' proteome followed a predestined pathway, regardless of sex and mode of delivery (Supplementary Fig. 11a, b). Correspondingly, the results of principal component analysis (PCA) (Supplementary Fig. 12a) and the related diffusion map (Supplementary Fig. 12b) both demonstrated similar results with the samples following a clear pattern based on time since birth.

Interestingly, the protein profile trajectory was most coherent right at birth and at full-term (Fig. 5b). Moreover, the most pronounced diversity in protein expression was observed at 1-week postnatal age. This suggests that the infants start life with similar protein profiles, followed by an interval where internal or external factors might be more influential before most infants converge their protein expressions again.

To examine the observed protein evolution in relation to neonatal immaturity, the UMAP result was investigated based on GA at birth as a proxy for fetal maturation. As seen in Fig. 5c, GA at delivery seemed to be of minor importance for protein expression at birth, as no distinction between GA groups can be seen on the first day of life. However, the degree of immaturity plays an increasing part in the differentiation, with a peak around 1 to 2 weeks after birth where the samples are clearly separated depending on GA at birth. Further on, the profiles converge once again as the infants grow older. From PMA 30 weeks, and

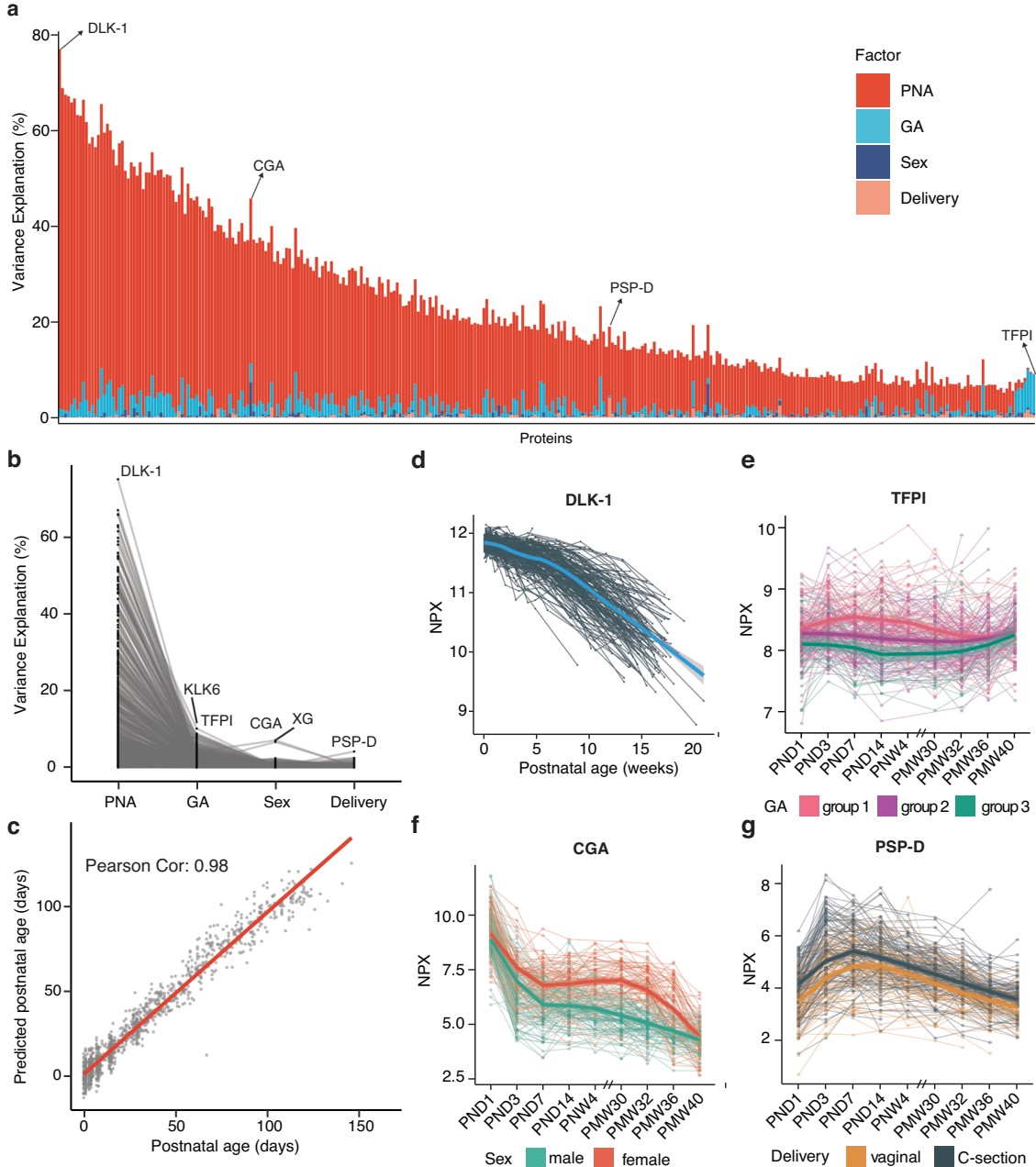

**Fig. 4 The variability of proteins explained by infant characteristics. a** Bar-plot showing the percentage of variance explanation of the factors postnatal age (PNA), gestational age (GA), sex, and delivery mode for all 538 analyzed proteins. PNA explains a clear majority of protein level variance. Specific proteins with other distinctive explanatory factors are highlighted. **b** Line plot showing the relative percentage of variance explanation of different factors for all analyzed proteins. Each line indicates a protein. **c** Correlations between the predictive PNA and chronological age for all 1335 samples using 151 proteins. **d–g** Longitudinal expression patterns of proteins over time, exemplified by the four proteins highlighted in Fig. 4a; DLK-1 (**d**), TFPI (**e**), CGA (**f**), and PSP-D (**g**). Each line indicates an individual. The regression lines were calculated based on samples analyzed from the same group.

especially 32 and persistently to 40 weeks PMA, no separation can be seen between GA groups. In addition, comparing the growth trajectories of infants in three GA groups, it was observed that those with smaller GA tended to have a slower rate of growth (Fig. 5d).

**Gestational effects on blood protein profiling.** The importance of GA on protein expression was additionally illustrated in Fig. 6a, where the numbers of proteins affected by the factors GA, sex, and delivery mode are presented per sampling time point. Consistent with Fig. 5c, GA at birth is most influential at 1-week of PNA, with a drastic decline at later postnatal ages.

To further explore how the GA group differs, the differentially expressed proteins (DEPs) between GA groups on PNA day 7 were analyzed by ANOVA (Supplementary Data 10) and visualized in a volcano plot (Fig. 6b). In total, 86 DEPs were identified, with some examples of proteins with decreased or increased levels in the infants with more advanced GA (Fig. 6c). The top 30 most significant DEPs were further analyzed in the radar plots (Fig. 6d and Supplementary Fig. 13). This analysis revealed the same pattern as seen in Figs. 5b and 6a. The three GA groups have similar protein profiles at birth, diverge into clearly varying trends at 1-week PNA but converge at full-term.

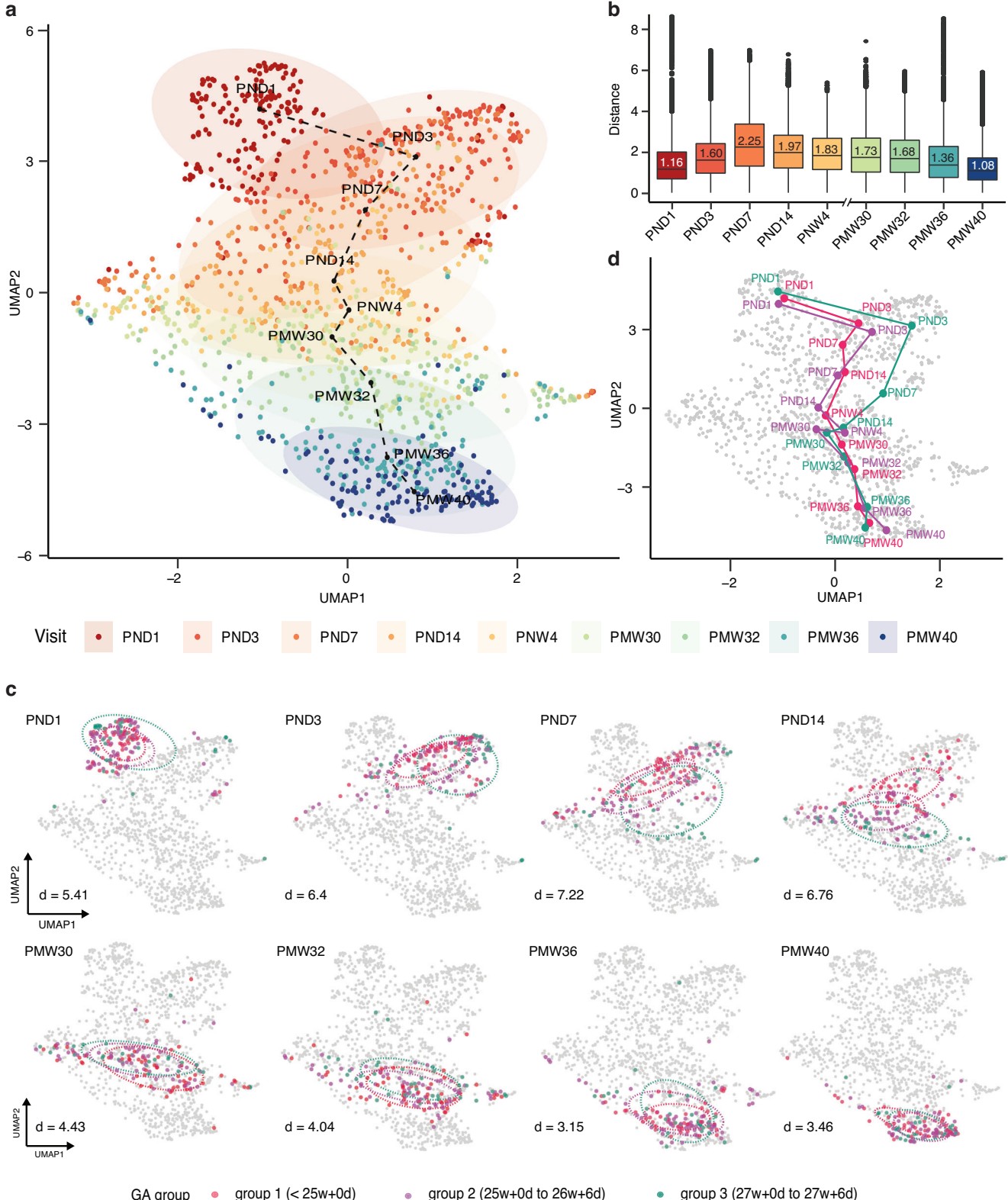

**Fig. 5 Longitudinal blood protein profiling of preterm infants during the postnatal period. a** UMAP clustering results based on 1335 samples presenting the dynamic changes in blood protein expression patterns of preterm infants after birth. The color code indicates the blood sampling time points. **b** Box-plot showing the distributions of Euclidean distance between the samples in at the sampling time points, whiskers show the largest and smallest values, color-coded by visit. Numbers within boxes represent the median distance. **c** UMAP clustering for preterm infants at each visit, color-coded by the three gestational age (GA) groups: group 1 (red), born at less than 25 + 0 (weeks + days) ($N = 61$); group 2 (purple), born at 25 + 0 to 26 + 6 (weeks + days) ($N = 81$); and group 3 (green), born at 27 + 0 to 27 + 6 (weeks + days) of gestation ($N = 40$). **d** Average protein evolution trajectories of infants in three GA groups.

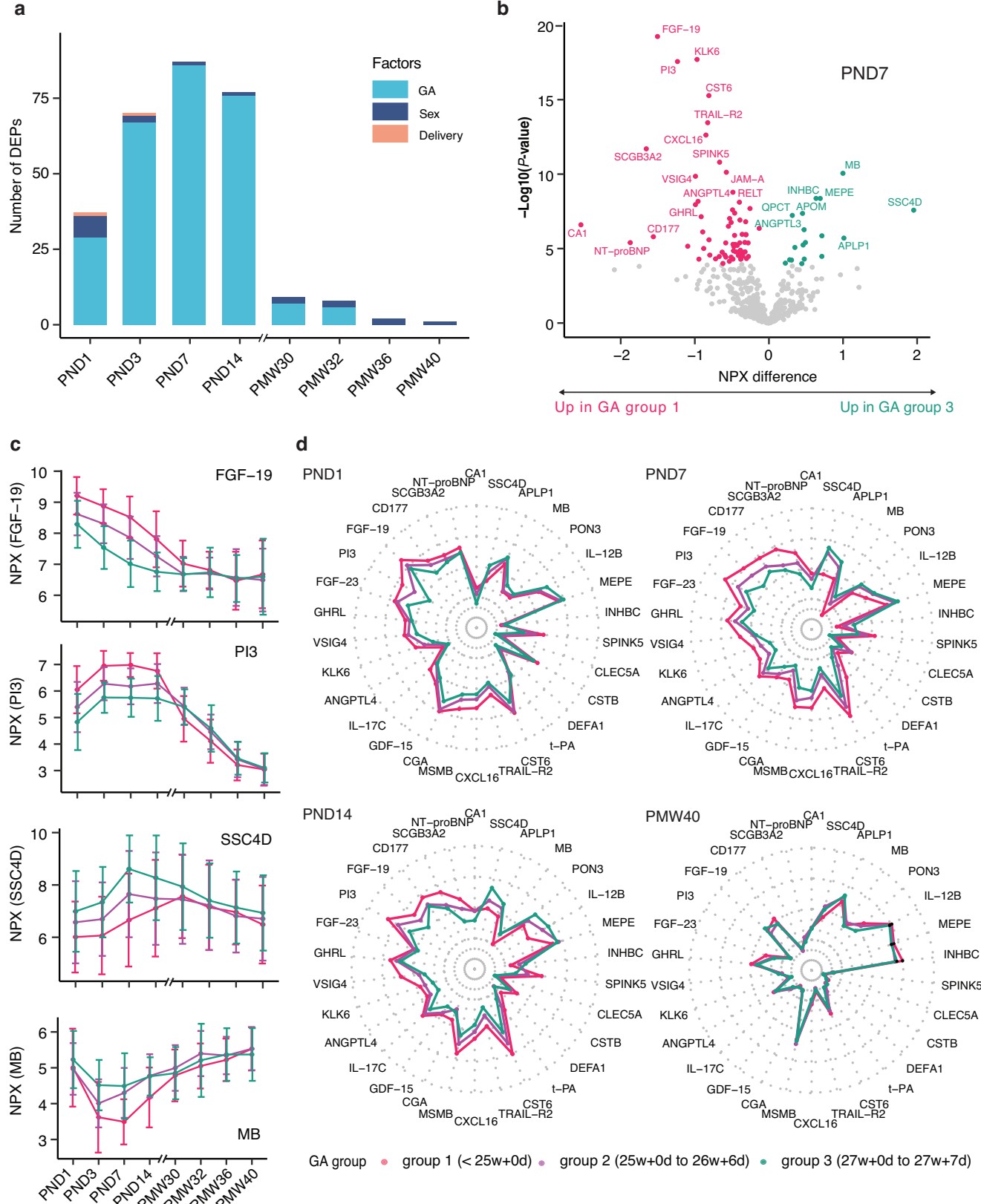

## Discussion

We report on a longitudinal study of postnatal blood protein profile development in extremely preterm infants in which "next generation blood proteome profiling" has been used. The fact that only a minimal blood volume is needed, without sacrificing accuracy or sensitivity, makes the assay platform ideal for analysis of such individuals. The combination of a large study cohort with an in-depth multiplex, high-throughput protein analysis platform, provides high-resolution insight into the complex physiology of preterm postnatal development. The study demonstrates a dynamic, stereotypic evolution of serum protein profiles from birth to full-term age. The pattern is consistent for the entire

**Fig. 6 Gestational age (GA) impact on blood protein expressions after preterm birth. a** Bar-plot representing the numbers of differentially expressed proteins related to GA, sex, and delivery mode. **b** Volcano plot showing the identified differentially expressed proteins ($n = 86$) across different GA groups on postnatal day 7. The x-axis represents log2 fold-change (FC) in GA group 3 compared to GA group 1. The y-axis represents −log10 adjusted P-values. Differentially expressed proteins were defined as proteins with adjusted P-values < 0.01 (three-way balanced ANOVA for GA with sex and delivery mode as covariates). Multiple test corrections have been applied to the P-values using the Benjamini and Hochberg method. **c** Four examples of up- and down-regulated proteins across three GA groups, error bars represent mean ± SD, color-coded by GA groups. GA group 1 (red), born at less than $25 + 0$ (weeks + days) ($N = 61$); GA group 2 (purple), born at $25 + 0$ to $26 + 6$ (weeks + days) ($N = 81$); and GA group 3 (green), born at $27 + 0$ to $27 + 6$ (weeks + days) of gestation ($N = 40$). **d** Radar plots showing median levels of top 30 differentially expressed proteins across three GA groups at PND1, PND7, PND14, and PMW40.

cohort, regardless of GA at birth, clinical differences, sex or growth. Protein variations were highly associated with postnatal age rather than any other factor, demonstrating the immense physiological impact of intrauterine-extrauterine transition. In addition, we show that the first weeks after birth is a period where protein expression differs most between individuals and the degree of immaturity seems to significantly impact the proteome, thus directing a window for potential diagnostics and therapeutics.

The fact that PNA is a more critical determinant than PMA for the development of the protein profiles might seem surprising, since in most neonatal clinical settings, the medical decisions, care and treatment are based on PMA rather than the PNA. Our findings thus suggest that it might be relevant to reevaluate this mindset and to include the aspect of PNA in the assessment of each patient. Our study suggests that the impact of GA at birth, a proxy used today for the degree of immaturity, is most prominent around 1 week after birth, but the effect of GA is heavily reduced after 30 weeks postmenstrual age. These observations are interesting since the infants born at lower GAs are more likely to be longer on parenteral nutrition, have a higher need for ventilator support and oxygen treatment, receive transfusions, and be subject to severe infections like sepsis[34]. One might speculate that the altered protein expression at 1 week of PNA may play a causal role, where the more immature infants may struggle to activate specific pathways for homeostasis, which leads to dysregulation and pathology. The reverse may also be true; several interventions such as transfusion with adult blood products, broad-spectrum antibiotics, and parenteral nutrition might influence the protein expression.

An analysis of the proteins with significantly higher expression in the most immature infants show that many of these proteins are involved in skin development, pulmonary maturation, myelinization, metabolism, and infection/inflammation. In contrast, the more mature group of infants have higher expression of proteins involved in osteogenesis, immune system maturation, and angiogenesis. This suggests that the more immature infants express proteins associated with reactive processes, such as inflammation, while the more mature infants express higher levels of proteins related to the development of the immune system.

A clear pattern of blood protein changes during the weeks after birth is evident in these preterm infants. Here, we have classified the most variable proteins into eight clusters, each showing a particular blood protein level trend during the weeks and months after birth, with five groups of proteins (clusters) displaying overall declining protein levels and three increasing trends after birth. Interestingly, many proteins with increased levels after birth have their origin in the liver, thus indicating the development of hepatic functions during the neonatal period. Similarly, the analysis of the proteins with origin from immune cells suggests that proteins from the dendritic cells increase after birth, while the expression of proteins from basophils and neutrophils decrease.

Overall, the current study has not found any strong correlation between protein levels in blood with sex or delivery mode. In most analyses, the sex had no or negligible effect on protein profiles. Nonetheless, a handful of proteins were subject to sex-related differences, exemplified with the lower level of protein CGA in male infants. Similarly, delivery mode was found to influence a limited number of proteins, of which the protein PSP-D showed the highest correlation between blood protein levels and delivery mode. Thus, cesarean-delivered infants had elevated levels of PSP-D as compared to vaginally delivered infants, which is interesting since PSP-D, a lung-specific protein that participates in the innate immune defense of the lungs, and this protein is considered a marker of lung injury. In this context, it is noteworthy that several earlier studies have demonstrated an increased risk for pulmonary disease after a cesarean section[37]. The data presented here thus supports earlier reports of associations between this protein and severe pediatric acute respiratory distress syndrome (PARDS) and bronchopulmonary dysplasia (BPD)[35,36]. Elevated levels of PSP-D have been described as indicating more severe PARDS, whereas low levels seem to predict worse pulmonary outcomes in BPD. Our findings support the suggestion that the administration of recombinant human PSP-D could be a promising therapy to prevent lung injury and BPD as suggested earlier[36].

This study has limitations. Although we examined the expression of more than 500 unique proteins, this only represents a fraction of the complete human blood proteome. Future studies should aim for more comprehensive protein coverage, which is now feasible even in minute amounts of blood through emerging platforms such as Olink Explore and SomeScan. In addition, 96% of the mothers received at least one dose of antenatal steroids and 79% received two doses. Magnesium sulfate was not introduced in clinical practice in Sweden at the time of the study. Hence, unfortunately, we cannot elucidate the impact of these interventions on the protein profile. Furthermore, additional information regarding disease etiology and development may be gained by analyzing the proteome in relation to maternal health status and complementary infant biochemical and genetic parameters, e.g., through metabolomics, transcriptomics, and microbiome analysis. We are now in the process of collecting such data, which will shed further light on preterm infants' adaptation to extrauterine life. Such analyses may also provide information on physiological processes occurring in tissues that are not necessarily captured within the serum proteome. Another limitation of our study is the lack of reference proteome data from full-term infants for comparison.

In conclusion, we report on a longitudinal study based on extremely preterm infants showing a consistent and stereotypic evolution of the blood protein profiles during the weeks and months after birth, mostly independent of GA, sex and mode of delivery. This comprehensive analysis of blood proteins contributes to filling the knowledge gap regarding the expression of proteins in extremely preterm infants in their transition from immaturity to term-equivalent age with clinical implications for guiding the diagnostic platform to improve the treatment regime of these infants.

**Data availability**

All summary statistics and association data are available in the Supplementary Data 1–10. Source data for Figs. 1, 2a, and 5 are available in Supplementary Data 11.

The proteomic data of the preterm infant cohort is available in the BioStudies database (http://www.ebi.ac.uk/biostudies) under accession number S-BSST843. Guidance on the analysis methods can be provided upon request by contacting the corresponding author.

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

## Acknowledgements

We thank the Plasma Profiling Facility at SciLifeLab in Stockholm for conducting the Olink analyses and the nurses at Sahlgrenska University Hospital, Karolinska University Hospital and Lund University Hospital for sample and data collection and Chatarina Löfqvist who contributed to the sample handling before the measurements. We acknowledge the entire Human Protein Atlas program staff and the Science for Life Laboratory for their valuable contributions. We would like to thank the Affinity Proteomics-Stockholm Facility at SciLifeLab for conducting the Olink proximity extension assay analyses. We also thank Chatarina Löfqvist for sample handling and Mattias Karlén for the illustrations. Funding was provided from the Erling Persson Foundation, the Swedish Medical Research Council #2016-01131, the Swedish Research Council #2022-01562, Government grants under the ALF agreements ALFGBG-717971 and ALFGBG-812951 and The Wallenberg Clinical Scholars, LillaBarnetsFond, Spädbarnsfonden (Swedish infant death foundation) and Region Stockholm (combined residency and PhD training program). This work was supported by the SciLifeLab & Wallenberg Data Driven Life Science Program (grant: KAW 2020.0239).

## Author contributions

W.Z., H.D., N.B., L.F., M.U., and A.H., designed and analyzed the data and wrote the manuscript. A.K.N. and U.S. were responsible for sample collection, handled the samples, and approved the manuscript. D.W., I.P., K.S., D.L., provided data, and advice, performed the clinical data collection, and approved the manuscript.

## Funding

## Competing interests

The authors declare no competing interests.
