## [Peer Review File · Communications Medicine]

The development of blood protein profiles in extremely preterm infants follows a stereotypic evolution patternReviewers' comments:

Reviewer #1 (Remarks to the Author):

Journal: Communications Medicine

Manuscript Number: COMMSMED-23-0029-T

Title: The development of blood protein profiles in extremely preterm infants follows a stereotypic evolution pattern

Summary

This is a longitudinal proteomic study on blood samples collected from the Mega Donna Mega study, a previous randomized control trial to study the effects of enteral fatty acid supplementations from birth to 40 weeks postmenstrual age (PMA) on retinopathy of prematurity (ROP).

The current study included 182 preterm infants born before 28 weeks of gestation. Serum samples were collected at nine time points from birth to term-equivalent age or 40 weeks PMA. The objectives of the study were to describe the changes in the protein profiles of extremely preterm infants and to relate the results to gestational age (GA), PMA, postnatal age (PNA), gender, and mode of delivery. The infants were classified into three groups: group 1, born at less than 25+0 (weeks+days) (N = 61); group 2, born at 25+0 to 26+6 (N = 81); and group 3, born at 27+0 to 27+6 weeks of gestation (N = 40). A total of 538 protein targets was measured by the proximity extension assay (PEA) technology including six Olink panels (cardiometabolic, cardiovascular II and III, development, inflammation, and metabolism) in 1335 serum samples.

The Authors found the following

1. Eight protein clusters with variable time-course patterns: five declining levels and three increasing levels with PMA were identified. Interestingly, the protein levels in the three GA groups were almost the same at birth, indicating similar protein expression patterns in different GA groups at birth.
2. Functional analysis of proteins in these clusters showed that most of the liver, lymphoid tissue, and salivary gland enriched proteins were increased after birth, indicating the development of hepatic

functions, and immune as well as metabolic shifts during the neonatal period. Many proteins that decreased after preterm birth were associated with the placenta, pancreas, and bone marrow.

3. A linear regression model suggested that the protein expression variations were primarily associated with PNA, while GA at birth is the second most explanatory factor. Neonatal gender and delivery mode impacted a few specific proteins and were substantially less influential compared to PNA.

4. Generalized linear models using 151 proteins were able to predict PNA with high accuracy (a Pearson correlation coefficient of 0.98).

5. Based on the longitudinal protein expression profile results, a stereotypic evolution of blood protein profiles from birth to term-equivalent age was described using Uniform Manifold Approximation and Projection (UMAP)

Interestingly, the protein profile trajectory of the three GA groups was most coherent at birth and at full term. Moreover, the most pronounced diversity in protein expression was observed at one week PNA. The Authors interpreted these findings as suggesting that infants start life with similar protein profiles, followed by an interval where internal or external factors might be more influential before most infants converge their protein expressions again. GA at birth is most influential at one week of postnatal age, with a drastic decline later.

6. To further explore how the GA group differs, the differentially expressed proteins (DEPs) between GA groups on PNA day 7 were analyzed by ANOVA and visualized in a volcano plot. In total, 86 DEPs were identified, with some examples of proteins with decreased or increased levels in infants with more advanced GA. The top 30 most significant DEPs were further analyzed in the radar plots.

The Authors concluded that there is a unified pattern of protein development for all infants, regardless of gestational age and clinical characteristics, suggesting an age-dependent (postnatal age) stereotypic development of blood proteins after birth. This knowledge should be considered in neonatal settings and might alter the clinical approach, where PMA or GA is currently the most dominant age variable.

Comment/Questions

This is an interesting study to the readers and other investigators with an important objective to describe the proteome of extremely preterm infants from birth to 40 weeks PMA. The large sample size, sophisticated analysis, and proteomic technology are strengths of this study. However, the study design and approach used in this study do not differentiate neonates with severe complications of prematurity and no evaluation of the effects of the intervention from the original RCT and the effects of the antepartum or postnatal management could be major drawbacks.

There are several questions and comments for the Authors as the following.

Major

Point#1. Since these neonates were randomized to enteral supplementation with arachidonic acid (AA) and docosahexaenoic acid (DHA) or placebo, did the Authors analyze the effects of these supplementations vs. placebo on the protein profiles? The original RCT (reference 19) showed the benefits of these supplementations in terms of reduction in the rate of severe ROP and had significantly higher fractions of AA and DHA in serum phospholipids compared with controls. In addition, the proteomes reported herein might not be applied to neonates who did not receive these supplementations.

Point#2. According to Supplementary data S1, there were several neonates who experienced complications of prematurity such as intraventricular hemorrhage, necrotizing enterocolitis, neonatal sepsis, and ROP. Could the Authors report the protein profiles of these neonates who had severe complications of prematurity compared to the profiles of neonates who did not have or had mild complications? Indeed, some of the Authors have published protein profiles of severe ROP from this dataset before. Why couldn't the Authors use the same approach with other complications of prematurity? The Authors proposed on Page 2, Line 64-65 that the knowledge about the postnatal proteome evolution in relation to PMA and PNA, respectively, can shed light on the role of early extrauterine development in health and disease.

Point#3. Are there any of the eight protein clusters associated with specific prematurity complications?

Point#4. Previous studies from the ELGAN Investigators demonstrated that blood protein profiles on the first postnatal day of infants born before 28 weeks differed by pregnancy complications (i.e. preeclampsia vs. spontaneous preterm labor or preterm PROM) (*Am J Obstet Gynecol* 2011;204:418.e1-12) and by placental histologic findings (i.e. inflammatory lesions vs. poorly perfused placental lesions) (*Pediatr Res* 2011; 69: 68–73). Could the Authors include these variables in addition to gender and mode of delivery on the linear regression models and variance analysis?

Point#5. Please describe the antepartum management of these neonates. Did the majority receive antenatal steroids to improve lung maturity or magnesium sulfate for neuroprotection? Could these medications make the protein profiles in the first 7 days of the three-GA group similar?

Point#6. According to Figure 2 A, the 9 time points for blood collection were PND1, PND3, PND7, PNW4, PMW30, PMW32, PMW 36, and PMW40. Was the sample collected on PNW4 mutually exclusive? If a neonate was born at 26 weeks, his PNW4 would be PMW 30 or if a neonate was born at 27 weeks, his PNW 4 would be PMW 31. How did the Authors solve this problem since additional blood samples would be collected on PMW 30 and PMW 32 too? The longitudinal profile graphs of each protein for the 9 time points might not be accurate. The Authors might have to add a broken line after PNW4 on the x-axis to indicate a separate scale.

Point #7. In variance explanation analysis, linear regression models suggested that the protein expression variations were primarily associated with PNA, while GA at birth is the second most explanatory factor. Did the Authors use the same unit for PNA and PMA (i.e. use days for both)? The coefficient and variance might be different if the report is the change per day vs. the change per week.

Point#8. The Authors claimed that a generalized linear model using 151 proteins was able to predict PNA with high accuracy (a Pearson correlation coefficient of 0.98). Could the Authors provide a Bland–Altman

plot which is a method of analyzing the agreement between two different measurements? How many percent of the calculated PNA were within +/- 3 days of the correct PNA?

Minor

Point #9. Please specify how much blood was obtained in each visit. How much volume of a serum sample is required for the proteomic PEA assays?

Point#10. Please provide the GA range in each group on Supplementary data1

Reviewer #2 (Remarks to the Author):

The authors have presented a very unique and insightful window into the proteome. The analyses were appropriate, and simplified to aid interpretation of the data. This unique longitudinal data set has provided several critical insights, namely in the role of PMA vs. PNA in its' association with the plasma proteome.

Below suggestions to improve the quality of the work.

-Proximity Extension Assay: Indicate that this was the Olink assay, and provide some citations of its' use in the literature, if any. If so, please indicate this in the introduction, as it is more familiar to the readership.

Introduction: you can also mention PMID 30862783, which also included proteomics in the first postnatal week among term newborns, and expand this to other developmental trajectories aside from the proteome, that may deliver similar biological insights (e.g. DOI: 10.1126/scitranslmed.adc9854).

Methods:

-Please provide more context for the Mega Donna Mega study. From which demographics (e.g. SES, ethnicity).

-Did any newborns become septic during the study?

-Were any data on infant feeding available (e.g. provision of breast vs. formula milk, any other parenteral feeds)? If so, were they incorporated into the analysis?

-Please comment on why PMA was used in place of Ultrasound dating. If ultrasound dating is available for some infants, please indicate how the GA assigned by Ultrasound compares to those determined by PMA compare.

Please provide details on blood collection and processing:

total volume of blood

total volume of serum collected, and the volume necessary for the protein assay.

Was blood refrigerated or kept at room temperature after collection?

What were the centrifugation conditions?

-Comment on any efforts to batch-randomize the samples across the plates by visit/treatment group/sex, any other key variables, and what informatic approaches were used to test and correct for batch effects.

-Data availability: consider also depositing analytical codes in a publicly-accessible repository.

Results

-Data S1: Please replace Gender with sex. Gender is not appropriately used here.

-Consider placing other social determinants in this table (e.g. SES, maternal education, household income, ethnicity, maternal age as some examples)

Figure 2:

-Consider performing longitudinal analysis by normalizing participants to themselves across time, to eliminate within-subject variation. This type of data transformation may improve the detection of developmental trajectories.

Figure 4: indicators for 4e, f, and g are missing.

Figure 5: The finding on profiles being more “coherent” at birth and term is interesting. Can that be quantified in some way, by, for example, indicating the variance at each time point separately? Likewise

Figure 5C is done very well. Another way to support this finding is to apply the PERMANOVA at each time point separately to indicate the variance explained by GA group within each PND time point.

Discussion:

-The authors have accurately described the implications of their results, without over-reaching. For the discussion pertaining to PNA being more relevant than PMA, this is the story that the data tells, but how does this observation correlate with clinical impression in the first few days of life? For example, there may be critical physiological processes that differ based on PNA that were not captured by the proteins measured in the serum, but may be within the tissues, perhaps captured by other types of omic modalities. The authors have made some comments in this respect, but these can be elaborated on.

The sex imbalance in recruitment is interesting. Does this reflect more males than females generally being admitted for care? If so, then this is more interesting given that there were minimal sex differences in the data. Can the authors include a direct comparison of males vs. females, or perform an analysis looking for effect modification? While minor, the differences may still be important, beyond the impact on CGA given in Figure 4.

-A limitations and future studies section should be included. This can include: any future comparisons to place the findings in context (e.g. comparison to term babies), any limitations (e.g. the SomaScan assay now has the ability to profile 7000+ proteins, and represents another powerful tool for discovery), and the next steps for this research

Reviewers' comments:**Reviewer #1:**

This is an interesting study to the readers and other investigators with an important objective to describe the proteome of extremely preterm infants from birth to 40 weeks PMA. The large sample size, sophisticated analysis, and proteomic technology are strengths of this study. However, the study design and approach used in this study do not differentiate neonates with severe complications of prematurity and no evaluation of the effects of the intervention from the original RCT and the effects of the antepartum or postnatal management could be major drawbacks.

There are several questions and comments for the Authors as the following:

Major

Point#1. Since these neonates were randomized to enteral supplementation with arachidonic acid (AA) and docosahexaenoic acid (DHA) or placebo, did the Authors analyze the effects of these supplementations vs. placebo on the protein profiles? The original RCT (reference 19) showed the benefits of these supplementations in terms of reduction in the rate of severe ROP and had significantly higher fractions of AA and DHA in serum phospholipids compared with controls. In addition, the proteomes reported herein might not be applied to neonates who did not receive these supplementations.

Authors: This is a relevant comment. We have investigated the effects of enteral supplementation with arachidonic acid (AA) and docosahexaenoic acid (DHA) compared to standard care on protein profiles by incorporating a 'fatty acid supplementation' factor in the linear regression model. While we observed associations between fatty acid treatment and protein expressions, these effects were generally less pronounced compared to PNA and GA. We have clarified this point in the main text (see line 205-207) and have included an assessment of the effects of fatty acid treatment in comparison to other factors in a new Supplementary Figure 7. As the primary scope of this study is focused on the associations between PNA, GA, and protein profiles, we are currently preparing a separate manuscript that will investigate infant protein levels in relation to measured serum concentrations of AA and DHA which will provide additional insights into the associations between fatty acids, fatty acid supplementation, and serum protein levels.

Point#2. According to Supplementary data S1, there were several neonates who experienced complications of prematurity such as intraventricular hemorrhage, necrotizing enterocolitis, neonatal sepsis, and ROP. Could the Authors report the protein profiles of these neonates who had severe complications of prematurity compared to the profiles of neonates who did not have or had mild complications? Indeed, some of the Authors have published protein profiles of severe ROP from this dataset before. Why couldn't the Authors use the same approach with other complications of prematurity? The Authors proposed on Page 2, Line 64-65 that the knowledge about the postnatal proteome evolution in relation to PMA and PNA, respectively, can shed light on the role of early extrauterine development in health and disease.

Authors: This is the first publication of blood protein analysis from this cohort, although we have previously published a smaller pilot study, from another intervention study supplementing parenteral DHA, on blood protein profiles of 14 preterm infants (PMID: 33895781 and PMID: 32330929). To address the reviewer's comment, we have added a 'morbidity' factor to our

linear regression model to investigate the effects of severe prematurity-related diseases on proteome expressions, see new Supplementary Figure 7. We identified associations between protein expressions and morbidities, but PNA and GA remain the most prominent factors on protein expressions. We plan to further explore these associations in depth, scrutinizing the medical files for various aspects, e.g. timing of disease onset, treatment of disease, and diagnostics of disease, in our follow-up studies and have added a sentence in the Discussion section to emphasize this point, see line 348-358.

Point#3. Are there any of the eight protein clusters associated with specific prematurity complications?

Authors: We appreciate the reviewer's comment regarding the potential associations between the eight protein clusters and prematurity complications. While we recognize the significance of this question, we have not yet performed these analyses in the current study. Our intention is to explore this matter in the future follow-up papers, as we want to collect detailed information about the diseases and their clinical manifestations before performing such analyses. We believe that a thorough and well-informed approach to these associations will provide more accurate results, see our answer above.

Point#4. Previous studies from the ELGAN Investigators demonstrated that blood protein profiles on the first postnatal day of infants born before 28 weeks differed by pregnancy complications (i.e. preeclampsia vs. spontaneous preterm labor or preterm PROM) (*Am J Obstet Gynecol* 2011;204:418.e1-12) and by placental histologic findings (i.e. inflammatory lesions vs. poorly perfused placental lesions) (*Pediatr Res* 2011; 69: 68–73). Could the Authors include these variables in addition to gender and mode of delivery on the linear regression models and variance analysis?

Authors: In response to the reviewer's suggestion, we have added 'preeclampsia' as an additional factor in the linear regression model and compared its effects on protein profiles alongside other factors, see line 205-207. We identified associations between preeclampsia and protein profiles; however, these effects were less prominent compared to the influences of PNA and GA. The results have been incorporated into a new Supplementary Figure 7. Spontaneous preterm labor and preterm PROM are not included in the present study. Placental tissue has been collected from two out of three study centers and the histology is currently in the process of being analyzed. This data will be presented together with more detailed maternal data in relation to the proteome, information which is also presently being collected for this purpose.

Point#5. Please describe the antepartum management of these neonates. Did the majority receive antenatal steroids to improve lung maturity or magnesium sulfate for neuroprotection? Could these medications make the protein profiles in the first 7 days of the three-GA group similar?

Authors: 96% of the mothers received at least one dose of antenatal steroids and 79% received two doses. Magnesium sulfate was not introduced in clinical practice in Sweden at the time of the study. Hence, unfortunately, we cannot elucidate the impact of these interventions on the protein profile. We have added information of antenatal steroid treatment in Supplementary data 1.

Point#6. According to Figure 2 A, the 9 time points for blood collection were PND1, PND3,

PND7, PNW4, PMW30, PMW32, PMW 36, and PMW40. Was the sample collected on PNW4 mutually exclusive? If a neonate was born at 26 weeks, his PNW4 would be PMW 30 or if a neonate was born at 27 weeks, his PNW 4 would be PMW 31. How did the Authors solve this problem since additional blood samples would be collected on PMW 30 and PMW 32 too? The longitudinal profile graphs of each protein for the 9 time points might not be accurate. The Authors might have to add a broken line after PNW4 on the x-axis to indicate a separate scale.

Authors: This is a relevant comment. To address this issue, continuous time scales have been used for statistical analyses when possible, for example when assessing the influence of PNA and PMA on protein profiles. To address the issue of overlapping time points, in instances when PNW 4 preceded PMW30, i.e., in infants born <26 weeks' GA, PNW4 was used. As such, PMW30 may include samples collected between just above 2 weeks postnatal age (infants born at GA 27 weeks) and 8 weeks postnatal age (infants born at GA 22 weeks). The distribution of the collected samples in relation to PNA and PMA is visualized in figure 1b. We have clarified how sample time points were defined in the Material and Methods, see line 457-460. Additionally, as suggested by the reviewer, we have added broken lines to all relevant figures (see changes in Figure 2a, 2e, 3b, 3b, 4d, 5b, 6a and 6c) to clarify the difference in scales.

Point #7. In variance explanation analysis, linear regression models suggested that the protein expression variations were primarily associated with PNA, while GA at birth is the second most explanatory factor. Did the Authors use the same unit for PNA and PMA (i.e. use days for both)? The coefficient and variance might be different if the report is the change per day vs. the change per week.

Authors: This is a relevant comment. We use the same unit (days) for PNA and PMA. We have added the information to the main text, see line 198-199.

Point#8. The Authors claimed that a generalized linear model using 151 proteins was able to predict PNA with high accuracy (a Pearson correlation coefficient of 0.98). Could the Authors provide a Bland–Altman plot which is a method of analyzing the agreement between two different measurements? How many percent of the calculated PNA were within +/- 3 days of the correct PNA?

Authors: As suggested by the reviewer, the discrepancies between predicted age and the actual chronological age have been present in a Bland-Altman plot as a new Supplementary figure 8. Approximately 40% (524 out of 1335 serum samples) of the calculated PNA were within +/-3 days of the correct PNA. However, 95 % of all samples were within the ± 1.96 SD range, indicating a good agreement between predicted and actual infant age. Interestingly, we observed that the differences between predicted age and actual chronological age (delta PNA) increased as the actual PNA increased. We also observed that for samples where the predicted age was lower than the actual chronological age, the infants tended to exhibit lower-than-normal weight gain (Supplementary Figure 9).

Minor

Point #9. Please specify how much blood was obtained in each visit. How much volume of a serum sample is required for the proteomic PEA assays?

Authors: Thank you for pointing out that this information was missing in the manuscript. 0.6 mL blood was collected at each visit and 1 μ L serum was used for each Olink panel. We have added this information to the Material and Methods section, see line 396 and 417.

Point#10. Please provide the GA range in each group on Supplementary data 1

Authors: The GA range for each group has been included in Supplementary data 1.

Reviewer #2:

The authors have presented a very unique and insightful window into the proteome. The analyses were appropriate, and simplified to aid interpretation of the data. This unique longitudinal data set has provided several critical insights, namely in the role of PMA vs. PNA in its' association with the plasma proteome.

Below suggestions to improve the quality of the work.

Reviewer: -Proximity Extension Assay: Indicate that this was the Olink assay, and provide some citations of its' use in the literature, if any. If so, please indicate this in the introduction, as it is more familiar to the readership.

Authors: As suggested by the reviewer, we have clarified in the introduction that the Proximity Extension Assay employed in our study is the Olink assay. Additionally, we have included references of Olink technologies and their applications in preterm research in the introduction, see line 71-74.

Reviewer: Introduction: you can also mention PMID 30862783, which also included proteomics in the first postnatal week among term newborns, and expand this to other developmental trajectories aside from the proteome, that may deliver similar biological insights (e.g. DOI: 10.1126/scitranslmed.adc9854).

Authors: As suggested by the reviewer, we have incorporated the suggested references in the introduction and expand the discussions to other developmental trajectories, see line 61-65 and 83-85.

Reviewer: Methods:-Please provide more context for the Mega Donna Mega study. From which demographics (e.g. SES, ethnicity).

Authors: The demographics of the Mega Donna Mega Study has been described in previous publication PMID: 33523106. The mothers had a mean age of 32 years. This has been added to Supplementary Data 1. Socioeconomic status and ethnicity of the mothers were not collected in the Mega Donna Mega Study.

-Did any newborns become septic during the study?

Authors: A total of 88 newborns experienced sepsis during the study. This information is provided in the Supplementary Data 1.

-Were any data on infant feeding available (e.g. provision of breast vs. formula milk, any other parenteral feeds)? If so, were they incorporated into the analysis?

Authors: We have detailed data on nutritional intake for the first weeks of infant life. Briefly, mother's own milk (when available) or donor milk was the only enteral feed until a postmenstrual age of 33 weeks. Thereafter, donor milk was replaced with preterm formula. The enteral feeds were introduced, if possible, from the first day of life. All infants received at least some parenteral nutrition in the neonatal period. We have added a reference to previous publications regarding the nutritional strategy in the Material and Methods section, see line 380-384. To evaluate the potential effects of mother's breast milk or donor milk on protein profiles, we incorporated a "percentage of mother's breast milk" factor into the linear regression model. The new analysis, which also includes comparisons with other factors, is present in a new Supplementary Figure 7. A sentence has been added to the main text to clarify this, see line 205-207.

-Please comment on why PMA was used in place of Ultrasound dating. If ultrasound dating is available for some infants, please indicate how the GA assigned by Ultrasound compares to those determined by PMA compare.

Authors: Ultrasound dating has been used for determination of gestational age for all patients. The term postmenstrual age (PMA) is here used for gestational age at birth + postnatal age. We have clarified this in the manuscript, see line 373.

Please provide details on blood collection and processing:

total volume of blood total volume of serum collected, and the volume necessary for the protein assay. Was blood refrigerated or kept at room temperature after collection? What were the centrifugation conditions?

Authors: We have added detailed information about sample collection and processing in the Material and Methods section (line 394-403 and 406-408).

-Comment on any efforts to batch-randomize the samples across the plates by visit/treatment group/sex, any other key variables, and what informatic approaches were used to test and correct for batch effects.

Authors: For this study, a total of 17 96-well-plates were used for the protein assay. To address potential batch effects, both internal and inter-plate controls are included. We also ensured that all longitudinal samples from one individual were allocated to the same plate to reduce batch effects related to inter-individual variability. Furthermore, a preterm infant serum pool sample and 8 internal control samples were included on each plate for bridging and quality control. We have incorporated these details in the Materials and Methods, see line 406-422.

-Data availability: consider also depositing analytical codes in a publicly-accessible repository.

Authors: We appreciate the reviewer's suggestion on the potential deposition of analytical codes. However, for this study, we did not generate any new or custom code. All of the analyses were conducted using standardized pipelines and existing tools described in the methods

section. Nonetheless, we are happy to provide guidance on the analysis methods if requested by other researchers. We have added this statement in the data availability section, see line 474-475.

Results

-Data S1: Please replace Gender with sex. Gender is not appropriately used here.

Authors: We have replaced the term ‘Gender’ with ‘Sex’ in Supplementary data 1.

-Consider placing other social determinants in this table (e.g. SES, maternal education, household income, ethnicity, maternal age as some examples)

Authors: The demographics of the Mega Donna Mega Study has been described in previous publication PMID: 33523106. The mothers had a mean age of 32 years, this has been added to Supplementary Data 1. Socioeconomic status, income, education level, and ethnicity of the mothers were not collected in the Mega Donna Mega Study.

Figure 2:

-Consider performing longitudinal analysis by normalizing participants to themselves across time, to eliminate within-subject variation. This type of data transformation may improve the detection of developmental trajectories.

Authors: This is a relevant comment. In our longitudinal analysis of protein expressions, we have considered sex, delivery mode, subject and GA at birth as confounding factors to eliminate within-subject variations as well as other potential confounding effects. We have added a sentence in the methods section to clarify this, see line 457-460.

Figure 4: indicators for 4e, f, and g are missing.

Authors: The indicators for 4e, f and g have been added.

Figure 5: The finding on profiles being more “coherent” at birth and term is interesting. Can that be quantified in some way, by, for example, indicating the variance at each time point separately? Likewise Figure 5C is done very well. Another way to support this finding is to apply the PERMANOVA at each time point separately to indicate the variance explained by GA group within each PND time point.

Authors: The distributions of distances between individuals at each visit are displayed in Figure 5b. Additionally, we have added the median distance values in the plot to provide a clearer representation of the variance at each time point.

Discussion:

-The authors have accurately described the implications of their results, without over-reaching. For the discussion pertaining to PNA being more relevant than PMA, this is the story that the data tells, but how does this observation correlate with clinical impression in the first few days of life? For example, there may be critical physiological processes that differ based on PNA that were not captured by the proteins measured in the serum, but may be within the tissues, perhaps captured by other types of omic modalities. The authors have made some comments in this respect, but these can be elaborated on.

Authors: This is a very good comment and clinically a very immature baby may certainly differ from a more mature baby during the early postnatal life course. However, surprisingly this did not come through in our analyses. We are now performing metabolomics, lipidomics and microbiome analyses in this cohort as well as scrutinizing the mothers' health status during pregnancy and analyzing placenta histology (when available). We believe this might help to understand what was not captured in the protein development analyses regarding the role of immaturity in postnatal development, see line 348-358.

The sex imbalance in recruitment is interesting. Does this reflect more males than females generally being admitted for care? If so, then this is more interesting given that there were minimal sex differences in the data. Can the authors include a direct comparison of males vs. females, or perform an analysis looking for effect modification? While minor, the differences may still be important, beyond the impact on CGA given in Figure 4.

Authors: This is correct, more males than females are born and admitted for care in these gestational age groups. We have now added a new volcano plot as Supplementary figure 6, which displays the differentially expressed proteins between males and females across the study visits.

-A limitations and future studies section should be included. This can include: any future comparisons to place the findings in context (e.g. comparison to term babies), any limitations (e.g. the SomaScan assay now has the ability to profile 7000+ proteins, and represents another powerful tool for discovery), and the next steps for this research

Authors: Thank you for this suggestion, we have now added a limitation paragraph that includes future perspectives, see line 348-358. So rightly pointed out it would have been optimal to compare our babies' protein development with full-term babies' development. We have now initiated such a study for future publications. We also aim to perform an additional analysis on a subset of our samples using the Olink explore platform covering more than 3000 proteins. One limiting factor is the availability of blood volumes in these tiny babies. Thus, most standard analytical approaches cannot be applied to this type of samples, with Olink, and possible SomaScan, being exceptions. As stated above, the next step within this research field is in this cohort to bring together and gather all the omics data, including maternal health status, to bring our research questions forward.

REVIEWERS' COMMENTS:

Reviewer #1 (Remarks to the Author):

Journal: Communications Medicine

Manuscript Number: COMMSMED-23-0029-T

Title: The development of blood protein profiles in extremely preterm infants follows a stereotypic evolution pattern

Comments to the Authors

The Authors have answered all questions from my previous comments. It is interesting to see that the major contributing factors to the changes in blood proteome in preterm neonates are postnatal age and perinatal morbidities make little difference in the first 7 days of life. All in-depth follow-up analyses regarding predictions of each adverse perinatal morbidity from blood proteome would be interesting.

Answer to Point #2: Could the Authors add the word “perinatal” to the term “morbidity” in the text and Supplementary Figure 7?

Answer to Point#5: The Authors answered that “96% of the mothers received at least one dose of antenatal steroids and 79% received two doses. Magnesium sulfate was not introduced in clinical practice in Sweden at the time of the study. Hence, unfortunately, we cannot elucidate the impact of these interventions on the protein profile.” Could the Authors add this information in the limitation section?

Answer to point #8: The Authors answered that “95 % of all samples were within the ± 1.96 SD range, indicating a good agreement between predicted and actual infant age.” Please add a parenthesis “(13 days)” to represent + 1.96 SD in the text (Line 212).

Reviewer #2 (Remarks to the Author):

The authors have addressed all my comments.

Reviewers' comments:

Reviewer #1:

Comments to the Authors

The Authors have answered all questions from my previous comments. It is interesting to see that the major contributing factors to the changes in blood proteome in preterm neonates are postnatal age and perinatal morbidities make little difference in the first 7 days of life. All in-depth follow-up analyses regarding predictions of each adverse perinatal morbidity from blood proteome would be interesting.

Answer to Point #2: Could the Authors add the word “perinatal” to the term “morbidities” in the text and Supplementary Figure 7?

Authors: We have added the word perinatal in the manuscript text as well as in the Supplemental Figure as suggested.

Answer to Point#5: The Authors answered and that “96% of the mothers received at least one dose of antenatal steroids 79% received two doses. Magnesium sulfate was not introduced in clinical practice in Sweden at the time of the study. Hence, unfortunately, we cannot elucidate the impact of these interventions on the protein profile.” Could the Authors add this information in the limitation section?

Authors: The information above has now been added to the limitation section in the manuscript.

Answer to point #8: The Authors answered that “95 % of all samples were within the ± 1.96 SD range, indicating a good agreement between predicted and actual infant age.” Please add a parenthesis “(13 days)” to represent + 1.96 SD in the text (Line 212).

Authors: The changes have been implemented as suggested.